# NAB2-STAT6 drives an EGR1-dependent neuroendocrine program in solitary fibrous tumors

Connor Hill[1,2], Alexandra Indeglia[1,3], Francis Picone[1], Maureen E Murphy[1], Cara Cipriano[4], Robert G Maki[4,5†], Alessandro Gardini[1]*

[1]The Wistar Institute, Philadelphia, United States; [2]Cell and Molecular Biology Graduate Group, Perelman School of Medicine, University of Pennsylvania, Philadelphia, United States; [3]Graduate Group in Biochemistry and Molecular Biophysics, Perelman School of Medicine, University of Pennsylvania, Philadelphia, United States; [4]Department of Orthopedic Surgery, University of Pennsylvania, Philadelphia, United States; [5]Abramson Cancer Center, Perelman School of Medicine, University of Pennsylvania, Philadelphia, United States

*For correspondence:
agardini@wistar.org

Present address: †Department of Medicine, Memorial Sloan-Kettering Cancer Center, and Weill Cornell Medical College, Cornell University, New York, United States

Competing interest: The authors declare that no competing interests exist.

## eLife Assessment

This study provides **compelling** data regarding the molecular characterization of a rare tumor type with few treatment options. This **fundamental** work significantly advances our mechanistic understanding of solitary fibrous tumours, a critical first step towards targeted precision medicine approaches. The results of this study will be of broad interest to cancer biologists and experimental oncologists.

**Abstract** The pathogenesis of many rare tumor types is poorly understood, preventing the design of effective treatments. Solitary fibrous tumors (SFTs) are neoplasms of mesenchymal origin that affect 1/1,000,000 individuals every year and are clinically assimilated to soft tissue sarcomas. SFTs can arise throughout the body and are usually managed surgically. However, 30–40% of SFTs will relapse local-regionally or metastasize. There are no systemic therapies with durable activity for malignant SFTs to date. The molecular hallmark of SFTs is a gene fusion between the *NAB2* and *STAT6* loci on chromosome 12, resulting in a chimeric protein of poorly characterized function called NAB2-STAT6. We use primary samples and an inducible cell model to discover that NAB2-STAT6 operates as a transcriptional coactivator for a specific set of enhancers and promoters that are normally targeted by the EGR1 transcription factor. In physiological conditions, NAB2 is primarily localized to the cytoplasm and only a small nuclear fraction is available to operate as a co-acti-vator of EGR1 targets. NAB2-STAT6 redirects NAB1, NAB2, and additional EGR1 to the nucleus and bolsters the expression of neuronal EGR1 targets. The STAT6 moiety of the fusion protein is a major driver of its nuclear localization and further contributes to NAB2's co-activating abilities. In primary tumors, NAB2-STAT6 activates a neuroendocrine gene signature that sets it apart from most sarcomas. These discoveries provide new insight into the pathogenesis of SFTs and reveal new targets with therapeutic potential.

## Introduction

The pathogenesis of many rare tumor types is poorly understood, preventing the design of effective treatments. For some of these tumors, the discovery of gene fusion events and the characterization of their biological repercussions has led to the design of effective treatments and improved patient outcomes. Notable examples include the MLL fusions that cause early onset mixed-lineage leukemias, the NUT-BRD4 fusions in NUT carcinoma, and the SS18 (SWI/SNF) fusions in synovial sarcoma (*French et al., 2003*; *Kadoch and Crabtree, 2013*).

SFTs are rare mesenchymal tumors with an estimated incidence rate of 1 in 1 million people per year (*Bansal et al., 2021*; *Gengler and Guillou, 2006*). They can develop in any location in the body, but most commonly arise in pleural, dural, or pelvic soft tissues. The majority of these sarcomas are indolent and can be surgically removed with curative intent. However, 30–40% of tumors recur local-regionally or metastasize and have no curative treatment options. Histology demonstrates that tumor masses present peculiar vascular features (hemangiopericytoma-like vessels) and are made up of patternless fibroblastic cells that are CD34+ (*Kazazian et al., 2022*). SFTs present no recurrent mutations at known oncogenes or tumor suppressors loci. However, an intrachromosomal inversion on chromosome 12 was identified as the molecular hallmark of SFTs in 2013 (*Robinson et al., 2013*; *Chmielecki et al., 2013*). The inversion on the long arm of chromosome 12 results in a gene fusion between *NAB2* and *STAT6*. Immunohistochemistry using antibodies targeting the C-terminus of STAT6 reveals strong nuclear staining in the presence of the chimeric protein NAB2-STAT6, leading to STAT6 nuclear staining as a diagnostic tool for SFTs (*Schweizer et al., 2013*). Hemangiopericytomas were previously diagnosed as a distinct soft tissue neoplasm and upon discovery of the *NAB2-STAT6* gene fusion, are now classified as SFTs (*Bansal et al., 2021*; *Gengler and Guillou, 2006*). Despite the introduction of molecular diagnostic tools, many SFTs are still misdiagnosed or misclassified, owing to our fundamental lack of understanding of their etiology.

The precise role of NAB2-STAT6 in SFTs pathogenesis remains unclear. Both NAB2 and STAT6 are physiologically active as transcriptional regulators. STAT6 is a DNA-binding transcription factor operating downstream to the JAK/STAT signaling pathway in leukocytes. STAT6 is commonly activated by cytokines such as IL-4 and IL-13 and mediates key immunological processes such as macrophage polarization and T-cell/B-cell activation (*Vahedi et al., 2012*; *Czimmerer et al., 2018*; *Shimoda et al., 1996*). Activation of STAT6 occurs via JAK-mediated phosphorylation at Y641, triggering homodimerization via the SH2 domain and ultimately leading to nuclear translocation (*Hou et al., 1994*; *Takeda et al., 1996*). Nuclear STAT6 stimulates transcription through its transactivation domain that recruits RNA Helicase A (RHA), SND1, and p300/CBP (*Litterst and Pfitzner, 2001*; *Wang et al., 2010*; *Yang et al., 2002*; *Välineva et al., 2005*).

Conversely, the NGFI-A binding protein 2 (NAB2) is expressed in most tissues and was originally characterized as a co-repressor of the Early Growth Response transcription factors (EGR1/2) (*Svaren et al., 1996*). The EGR proteins are immediate early genes (IERs), like other ubiquitously expressed transcription factors such as AP-1 (all FOS/JUN family members) (*Weaver et al., 2007*; *Coleman et al., 1992*). Immediate Early Genes are upregulated in most cell types in response to a variety of growth stimuli (*Healy et al., 2013*; *Gallo et al., 2018*). In fact, EGR1/2 are considered, like AP-1, broad regulators of cell proliferation and peak during developmental processes (*Thiel and Cibelli, 2002*). Their role has been best studied in the central nervous system (CNS), where they also regulate neuronal activity post-mitosis (*Giudicelli et al., 2001*; *Poirier et al., 2008*; *Thierion et al., 2017*). EGR1 can modulate the expression of growth factors, such as IGF2 and TGF-β1, as well as that of NAB2, and is believed to act either as an oncogene or tumor suppressor depending on the cellular context (*Saha et al., 2021*; *Wang et al., 2021*; *Bae et al., 1999*; *de Belle et al., 1999*). NAB2 forms homodimers and heterodimerizes with its family member NAB1, which was also described as a co-repressor of EGR1-dependent transcription (*Svaren et al., 1998*). We recently demonstrated that NAB2 may also function as co-activator of transcriptional enhancers during myeloid differentiation (*Barbieri et al., 2018*). The global role of NAB2 as a chromatin regulator in tumors, either in its wild-type form or as a fusion protein partner, remains poorly elucidated.

Several different NAB2-STAT6 isoforms have been identified. Most tumors (40–70%, especially those of pleural origin) carry a large fusion product of ~140 Kda resulting from exons 1–4 of NAB2 (retaining a truncated NCD2 domain and missing the C-terminal CID domain) and exons 2–22 of STAT6 (with the entire CDS). Another frequent isoform retains a larger NAB2 moiety (exon 1-5/6, to

include a truncated CID domain) and a much smaller STAT6 moiety (exon 16/17–22, missing the SH2 and DNA binding domains but retaining the transactivation domain). This shorter isoform accounts for 10–20% of cases and is most common in SFTs originating in extra-pleural locations. Risk of malignant progression, recurrence, and metastasis are not preferentially associated with either isoform of NAB2-STAT6 (*Bieg et al., 2021*; *Park et al., 2019*; *Tai et al., 2015*; *Yuzawa et al., 2016*; *Chuang et al., 2016*). In fact, the current risk assessment model (Demicco score) does not incorporate any molecular or genetic features of the primary tumor (*Demicco et al., 2012*).

The determinants of NAB2-STAT6 occupancy genome-wide have not been previously investigated, nor have the epigenetic and transcriptional consequences of the fusion protein been identified (*Robinson et al., 2013*; *Park et al., 2019*; *Park et al., 2020*; *Li et al., 2023*). Here, we employ functional genomics and proteomics strategies to demonstrate that NAB2-STAT6 drives a EGR1-dependent neuroendocrine transcriptional program via epigenetic activation of a network of distal enhancers and proximal promoters that is normally operating in neuronal cells. We further show that NAB2 is primarily contained in the cytoplasm and becomes nuclear-bound when fused to STAT6. The fusion protein utilizes the full EGR1 regulatory axis, including homo- and heterodimerization with NAB1/NAB2, to drive a highly distinctive transcriptional program that sets SFTs apart from most other tumors of mesenchymal origin.

## Results

### Solitary fibrous tumors express a neuronal gene signature

Developing treatments for SFTs is particularly challenging since their molecular landscape is largely undefined. Whole-exome sequencing of primary tumors has led to the discovery of *NAB2-STAT6* gene fusions. Additional transcriptomic data have been previously obtained, but were difficult to interpret in the absence of matching normal tissues (*Robinson et al., 2013*; *Georgiesh et al., 2021*). We examined formalin-fixed paraffin-embedded (FFPE) sections of eight tumors and their normal tissue counterpart. All tumors were removed from the pleura or pleural-adjacent locations such as lung, chest wall, and hilum. Three of these tumors were diagnosed as malignant, three as low risk or benign, and the other two had uncertain biological potential (*Figure 1a*). We extracted RNA from all FFPE sections and subjected it to exon-targeted sequencing (*Figure 1b*). To validate SFT diagnosis for each tumor, we employed Arriba, a computational tool to identify gene fusions from RNA-seq data. All tumors shared one common NAB2-STAT6 isoform, spanning the first four exons of NAB2 and all but the first STAT6 exon (*Figure 1c*, *Figure 1—figure supplement 1a*). The transcriptomes of the primary SFTs well correlated with previously published SFT RNA-seq datasets (*Figure 1—figure supplement 1b*). We performed differential gene expression analysis between the tumors and the adjacent normal tissues (*Figure 1—figure supplement 1c*). We found 2429 genes significantly upregulated across all SFTs. Notably, the most upregulated gene was *IGF2*, an EGR1 target that was recognized as a major driver of a hypoglycemic condition occurring in <5% of SFTs (Doege-Potter syndrome). We further identified key regulators of neuronal development as upregulated, including *SHOX2, KCNA1, LHX2*, and *ROBO2*. A set of upregulated genes encoded GABA receptor subunits (*GABRD, GABRA2, GABRB2, GABRG1*). We also identified 3769 downregulated genes, including a set of immune processes regulators such as *CCL18, IGHA1, SCGB3A1, IGHA2,* and *PIGR* (*Figure 1d* and *Supplementary file 1*). Gene ontology and gene set enrichment analysis (GSEA) revealed that neuronal development programs are broadly upregulated in SFTs (*Figure 1e*, *Figure 1—figure supplement 1e*). Conversely, immune and cell signaling pathways are significantly downregulated (*Figure 1—figure supplement 1d–e*), suggesting that SFTs may be immunologically 'cold'. We analyzed the enrichment of transcription factor motifs within the promoter regions of all upregulated genes identified by RNA-seq. We found that binding sites for DB1, MAZ, MOVO−B, and EGR1 were the top enriched motifs by adjusted p-value and that EGR1 motifs were over-represented in the whole enrichment matrix (*Figure 1f*). Conversely, the top motifs enriched at the promoters of downregulated genes belonged to transcription factors active in mesoderm-derived tissues, like HTF4 and MRF4 (*Figure 1—figure supplement 1f*). These results suggest that an EGR1-bound neuronal transcriptional program is being activated in SFT cells.

### Generation of an inducible NAB2-STAT6 cell model

SFTs are presumed to be mesenchymal in origin, therefore, we used the osteosarcoma-derived U2OS cell line to generate an inducible model of NAB2-STAT6 expression and study the early transcriptional



**Figure 1.** Solitary fibrous tumors express a neuronal gene signature. (**a**) Table of the eight formalin-fixed paraffin-embedded (FFPE) samples used for RNA-seq listing the sample ID, Age at surgery, sex, diagnosis of tumor status, and site of resection. (**b**) Graphic demonstrating the workflow for FFPE RNA-seq. RNA was extracted from FFPE tumor and matching normal tissue and sequenced using exon-targeted sequencing which increases quality of FFPE RNA-seq's. (**c**) Most abundant gene fusion present in all solitary fibrous tumors (SFTs) in FFPE RNA-seq, NAB2-STAT6 with exons 1–4 of NAB2

*Figure 1 continued on next page*

Figure 1 continued

and exons 2–22 of STAT6. Graphic generated with Arriba, a fusion detection algorithm. (**d**) Volcano plot showing differentially expressed genes in SFTs versus normal matching tissues as determined by FFPE RNA-seq (n=8). 2429 genes were upregulated (indicated by red dots) and 3769 genes were downregulated (indicated by blue dots). Fold change >1, FDR <0.1. (**e**) Biological pathway gene ontology (GO) analysis of 2429 upregulated genes in INTS10 KO cells revealed enrichment for developmental and specifically neuronal developmental pathways. (**f**) TRANSFAC motif (transcription factor motifs at +/-1 kb from transcription start site) GO analysis of 2429 upregulated genes in INTS10 KO cells revealed enrichment for DB1, MAZ, MOVO-B, EGR1, and WT1 motifs.

The online version of this article includes the following figure supplement(s) for figure 1:

**Figure supplement 1.** Solitary fibrous tumors express a neuronal gene signature.

events driven by the fusion protein. Previous work on NAB2-STAT6 relied on overexpression in bulk cell populations that were primarily of non-mesenchymal origin, potentially limiting their utility (*Robinson et al., 2013*; *Park et al., 2019*; *Park et al., 2020*; *Li et al., 2023*). We generated a single-cell-derived clone expressing the most common isoform of NAB2-STAT6 (exons 1–4 of *NAB2* and exons 2–22 of *STAT6*) under the control of a doxycycline-inducible promoter (tet-ON). We observed expression of NAB2-STAT6 starting at 24 hr of doxycycline treatment, peaking at 48 hr (*Figure 2a*). To profile early gene expression changes induced by the expression of NAB2-STAT6 we performed 3' mRNA QuantSeq. We observed widespread gene dysregulation that increased in magnitude over time, indicating that the fusion protein has direct and robust effects genome-wide (*Figure 2b*, *Figure 2—figure supplement 1a*). We observed a smaller cluster of 299 genes exhibiting the strongest upregulation after 1 day of NAB2-STAT6 expression (cluster 1) and tapering off at 48 hr and 72 hr of doxycycline (*Figure 2b*), suggesting they could be indirect targets. This cluster was enriched for translation and protein biosynthesis genes that are regulated by E2F, ZF5, and HES-7 transcription factors (*Figure 2—figure supplement 1b–c*), further pointing to an immediate cellular stress response to the ectopic expression of NAB2-STAT6. Larger scale gene expression changes were apparent after 48 hr of doxycycline with 562 upregulated genes (73%), including IGF2 and several neuronal regulators. About 27% of differentially expressed genes were downregulated (*Figure 2c* and *Supplementary file 2*), indicating that the primary function of NAB2-STAT6 may be transcriptional activation. Upregulated pathways were primarily involved in neuronal differentiation and development (*Figure 2d*). Promoters of upregulated genes were overwhelmingly enriched for EGR1 motifs, while STAT motifs were not significantly enriched (*Figure 2e*). The expression of several neuronal markers, such as *LHX2, ROBO2, SHOX2* increased over time of doxycycline treatment and peaked at 72 hr of NAB2-STAT6 expression (*Figure 2f*). Interestingly, the expression of endogenous *NAB2, NAB1,* and *EGR1* followed a similar trend, suggesting that the fusion protein enables a feed-forward mechanism of the EGR1 regulatory axis (*Figure 2f*). NAB2 has been proposed to repress its own expression and that of *NAB1* and *EGR1*. However, this trend suggests that NAB2-STAT6 does not possess any co-repressor activity and is instead functioning as a co-activator.

The GO and TF enrichment analyses in U2OS cells closely matched our findings from primary SFTs (*Figure 1*), despite a limited overlap between the gene lists (15%, *Figure 2—figure supplement 1d*). Key EGR-1 targets such as *IGF2* or the neuronal development regulators *LHX2* and *ROBO2* were commonly upregulated, indicating that U2OS cells can be utilized to model the activity of the fusion protein but do not accurately represent the cell of origin of SFTs (*Figure 2—figure supplement 1d*).

## EGR1 targeted promoters and enhancers are activated by NAB2-STAT6

We established that EGR1 neuronal targets are upregulated in SFTs and after NAB2-STAT6 expression in U2OS cells. To investigate the mechanistic underpinnings of NAB2-STAT6 dysregulation, we performed ChIP-seq of the fusion protein through its C-terminal FLAG epitope after 48 hr of doxycycline. We identified 1394 peaks that gained significant NAB2-STAT6 signal compared to day 0 and were almost equally distributed between promoters (57%) and distal cis-regulatory regions (43%, *Figure 3a*, *Figure 3—figure supplement 1a*, and *Supplementary file 3*). Next, we analyzed endogenous EGR1 at NAB2-STAT6 sites and found seeding binding across the vast majority of fusion protein peaks (*Figure 3a*). Upon NAB2-STAT6 expression, EGR1 binding increased at least twofold at most sites. We performed ATAC-seq to investigate chromatin accessibility status. NAB2-STAT6 peaks were moderately accessible at day 0 and their accessibility increased twofold (similar to EGR1 binding) at 48 hr of doxycycline (*Figure 3a*). We next profiled endogenous NAB2 and STAT6 in control conditions

**Figure 2.** Generation of an inducible NAB2-STAT6 system to investigate early transcriptional changes. (**a**) We generated a doxycycline-inducible clone that expresses NAB2-STAT6 (NAB2 exons 1–4, STAT6 exons 2–22) with a C-terminal FLAG tag. Immunoblot analysis of whole cell extracts shows strong expression of NAB2-STAT6 after 1, 2, and 3 days of doxycycline treatment using a FLAG antibody. GAPDH was used as control. (**b**) Heatmap clustering analysis of 2430 genes that are differentially expressed (fold change >1, FDR <0.1) across 1, 2, and 3 days of NAB2-STAT6 expression (Dox) as

*Figure 2 continued on next page*

*Figure 2 continued*

determined by 3' mRNA Quant-seq (n=4) (**c**) Volcano plot showing differentially expressed genes in cells expressing NAB2-STAT6 (Dox) for 2 days versus control cells as determined by 3' mRNA Quant-seq (n=4). 562 genes were upregulated (indicated by red dots) and 211 genes were downregulated (indicated by blue dots). Fold change >1, FDR <0.1. (**e**) Biological pathway GO analysis of 562 upregulated genes in INTS10 KO cells revealed enrichment for neuronal developmental pathways. (**e**) TRANSFAC motif (transcription factor motifs at +/-1 kb from transcription start site) gene ontology (GO) analysis of 562 upregulated genes in INTS10 KO cells revealed enrichment for EGR1 and PATZ motifs. (**f**) We plotted normalized read counts of NAB1, NAB2, EGR1, EGR2, STAT6, EGR target IGF2, and neuronal markers m (LHX2, ROBO2, and SHOX2) over 3 days of NAB2-STAT6 (Dox) expression. Targets of EGR1 were gradually upregulated; *NAB1, NAB2, EGR1, IGF2, LHX2, ROBO2,* and *SHOX2*.

The online version of this article includes the following source data and figure supplement(s) for figure 2:

**Source data 1.** Original scans with captions.

**Source data 2.** Raw immunoblot scans for *Figure 2*.

**Figure supplement 1.** Generation of an inducible NAB2-STAT6 system to investigate early transcriptional changes.

and found that NAB2 localized at NAB2-STAT6 peaks but not STAT6 (*Figure 3—figure supplement 1b*). Conversely, NAB2-STAT6 did not bind any STAT6 control sites (*Figure 3—figure supplement 1c*). To further assess determinants of NAB2-STAT6 binding, we performed motif analysis using HOMER and identified a strong EGR1/EGR2 signature but no STAT family motifs (*Figure 3b*). Gene set enrichment analysis (GSEA) showed that NAB2-STAT6 target genes, as determined by ChIP-seq, were highly enriched in NAB2-STAT6 upregulated genes found by RNA-seq analysis (*Figure 3c*). NAB2-STAT6 and EGR1 co-localize at promoters and enhancers of neuronal markers such as KNDC and UNCX (*Figure 3d*, *Figure 3—figure supplement 1d*). Additional enhancer sites targeted by NAB2-STAT6 were found near known EGR1 target genes such as IGF2, the most upregulated gene in SFTs (*Figure 3e*). Interestingly, the fusion protein also targeted the proximal promoter of EGR1, thereby boosting its expression and providing a robust feed-forward mechanism to sustain its own epigenetic network (*Figure 3—figure supplement 1e*). Collectively, these results demonstrate that NAB2-STAT6 localizes to EGR1 targets increasing their accessibility and expression.

## NAB2-STAT6 localizes to EGR1 targets in primary tumors

To validate our U2OS model of NAB2-STAT6 as a robust activator of certain EGR1 targets, we set out to determine genome-wide NAB2-STAT6 occupancy in a primary solitary fibrous tumor. We obtained a fresh primary pre-sacral SFT post-surgery (no radiation or chemotherapy had been administered to the patient). This tumor expressed the short isoform of NAB2-STAT6 containing exons 1–6 of NAB2 and exons 17–22 of STAT6 (*Figure 4—figure supplement 1a*). We isolated and fixed in single cell suspension ~50 million cells and performed ChIP-seq for NAB2, STAT6, and RNA Polymerase II (RNAPII). Since we previously showed that endogenous NAB2 and STAT6 localization do not overlap in physiological conditions (*Figure 4—figure supplement 1b*), we reasoned that we could pinpoint NAB2 and STAT6 primary binding sites as well as specific binding sites of the fusion protein, based on the convergence of NAB2 and STAT6 signals (*Figure 4a*). We identified 6284 NAB2 peaks, 1640 STAT6 peaks, and 38,036 RNAPII peaks in primary tumor cells. Peak overlap analysis revealed 718 putative NAB2-STAT6 binding sites, 69% of which were distal to gene promoters and likely to represent transcriptional enhancers (*Figure 4b*, *Figure 4—figure supplement 1c* and *Supplementary file 4*). The NAB2-STAT6 peaks also showed robust RNAPII signal, further suggesting that the fusion protein elicits transcriptional activation (*Figure 4b*). We also identified 5912 unique NAB2 peaks and 1285 unique STAT6 peaks (*Figure 4b*). Next, we performed motif analysis using HOMER. NAB2 only sites were enriched for EGR1/2 motifs (*Figure 4c*, *Figure 4—figure supplement 1d*) and the same motif profile was found at NAB2-STAT6 peaks (*Figure 4c*, *Figure 4—figure supplement 1e*). STAT6 only peaks, instead, were enriched for GRE and STAT motifs (*Figure 4c*, *Figure 4—figure supplement 1f*). Gene set enrichment analysis of NAB2-STAT6 peaks (closest gene) showed significant correlation with upregulated genes in SFTs (*Figure 4d*). Collectively, these findings align with data obtained in U2OS cells and suggest that NAB2-STAT6 localization is exclusively driven by EGR1 binding. We found limited overlap between NAB2-STAT6 sites in the primary tumor and those retrieved in U2OS (*Figure 4e-f*, *Figure 4—figure supplement 1g-h*), consistent with the limited overlap between their transcriptomes. It is likely that the mechanism of transcription activation by NAB2-STAT6 is conserved, whereas the targets are cell-type specific. Interestingly, NAB2 appeared upregulated in both systems as either the fusion protein or endogenous NAB2 bound the proximal promoter region (*Figure 4f*),



**Figure 3.** EGR1 targeted promoters and enhancers are activated by NAB2-STAT6. (**a**) Average profiles and heatmaps of NAB2-STAT6 FLAG and EGR1 ChIP-seq and ATAC-seq in both control and 2 days NAB2-STAT6 (Dox) expressing U2OS cells at 1394 NAB2-STAT6 FLAG peaks. NAB2-STAT6 FLAG becomes significantly localized to these peaks which have significant increases in EGR1 and ATAC-seq signal. (**b**) Motif analysis of 1394 NAB2-STAT6 FLAG peaks using HOMER shows EGR1, EGR2, and WT1 as the most significantly enriched TF matrices. (**c**) Gene set enrichment analysis (GSEA) shows

*Figure 3 continued on next page*

Figure 3 continued

that genes nearest to NAB2-STAT6 FLAG peaks (n=1394) are significantly upregulated after 2 days of NAB2-STAT6 (Dox) expression in U2OS cells when compared with control cells from *Figure 2*. (**d**) Screenshot displays two enhancers and the promoter (highlighted in yellow) of *KNDC1* that gains NAB2-STAT6 FLAG localization and has increases in EGR1 localization and accessibility by ATAC-seq. (**e**) Screenshot displays an enhancer (highlighted in yellow) of *IGF2* that gains NAB2-STAT6 FLAG localization and has increased EGR1 localization and accessibility by ATAC-seq.

The online version of this article includes the following figure supplement(s) for figure 3:

**Figure supplement 1.** EGR1 targeted promoters and enhancers are activated by NAB2-STAT6.

furthermore, nucleosome accessibility and EGR1 binding increased in U2OS cells upon doxycycline treatment (*Figure 4f*). The oncogenic NAB2-STAT6 fusion may thereby reinforce its own expression in SFTs.

## The subcellular localization and interactome of NAB-STAT6

Previous work established that NAB2-STAT6 localizes to the nucleus. It has been proposed that NAB2 is primarily guiding the subcellular localization of the chimeric protein, barely contributing any further functional domain, whereas STAT6 endows the fusion product with activation domain (*Schweizer et al., 2013*). Since the interactome of NAB2-STAT6 has not been previously investigated with unbiased proteomic approaches, we performed LC-MS/MS analysis on NAB2-STAT6 eluates using a MudPIT approach. First, we affinity-purified NAB2-STAT6 from nuclear fractions of U2OS cells using antibodies directed against NAB2 or STAT6. Both antibodies efficiently immunoprecipitated the chimeric protein, as well as NAB1 and EGR1 (*Figure 5a*). To validate EGR1 as a NAB2-STAT6 interactor, we performed the reciprocal IP. Upon doxycycline induction in U2OS, EGR1 antibodies co-precipitated NAB2-STAT6. In addition, EGR1 co-purified along with CBP/P300 and RNAPII subunits suggesting that NAB2-STAT6 co-opts EGR1 co-activator functions (*Figure 5—figure supplement 1a*). To further validate NAB2-STAT6's interactome, we performed FLAG affinity purification in U2OS as well as in a 293T stably expressing clone (*Figure 5—figure supplement 1b*). This approach pulled down the fusion product as well as near-stoichiometric amounts of NAB1, additional interactors included the co-activators SND1 (normally recruited by STAT6) and RHA (recruited by either STAT6 or EGR1, *Figure 5—figure supplement 1c–d*). NAB1 is a NAB2 paralog that was loosely studied for its ability to repress EGR1-mediated activation. NAB1 is the strongest interactor of NAB2-STAT6 across multiple experiments (*Figure 5—figure supplement 1d*), suggesting that a heterodimeric form of the oncogenic protein may be its most common form. Endogenous NAB2 can also heterodimerize with NAB1, as we have shown (*Figure 5—figure supplement 1d*). Additionally, the chimeric protein can heterodimerize with endogenous, full-length NAB2 (*Figure 5—figure supplement 1e–f*).

To further clarify the subcellular localization of NAB2-STAT6, we initially performed fractionation experiments in U2OS cells to find that NAB2 and EGR1 localization is primarily cytoplasmic prior to NAB2-STAT6 expression, whereas endogenous STAT6 is nuclear (*Figure 5b*). The fusion protein is fully retained in the nucleus and drives endogenous NAB2 and EGR1 to the nucleus (*Figure 5b*). To corroborate these findings, we employed immunocytochemistry (ICC). We first examined the primary tumor cells that we profiled by ChIP-seq (*Figure 4*) and retrieved a robust nuclear signal for NAB2 and STAT6 (*Figure 5c*). Next, we performed ICC in the U2OS inducible clone and confirmed that in wild-type conditions, NAB2 localization is largely cytoplasmic, while STAT6 signal is mostly nuclear (*Figure 5c*, *Figure 5—figure supplement 1g*). Upon NAB2-STAT6 expression, NAB2 and STAT6 signals co-localize in the nucleus, while a small amount of endogenous NAB2 remains cytoplasmic. Taken together, these data suggest that the STAT6 moiety drives nuclear localization of the fusion protein. The NAB2 paralog NAB1 heterodimerizes with the fusion protein (*Figure 5a*, *Figure 5—figure supplement 1d*). Accordingly, NAB1 under physiological conditions is predominantly cytoplasmic (*Figure 5d*, *Figure 5—figure supplement 1h*) and is redirected to the nucleus by NAB2-STAT6 (*Figure 5d*), having a similar fate to that of endogenous NAB2 (*Figure 5b*). Importantly, NAB1 antibodies do not cross-recognize NAB2, and vice versa (see Author response to Reviewers).

## The SFT gene signature is expressed in neuroendocrine tumors

SFTs are traditionally assimilated to soft tissue sarcomas and are often positive to CD34 staining. They are broadly classified as mesenchymal tumors based on their histological patterns, albeit their cell of origin remains uncertain (*Jo and Demicco, 2022*). To unbiasedly determine which tumor



**Figure 4.** NAB2-STAT6 localizes to EGR1 targets in primary tumors. (**a**) Summary of the strategy to profile NAB2-STAT6 binding in a primary solitary fibrous tumor (SFT). The primary tumor was designated and single cells were isolated and fixed. Fixed cells were then used for NAB2 and STAT6 ChIP-seq. Peaks overlapping in both NAB2 and STAT6 ChIP-seq were characterized as NAB2-STAT6 peaks. (**b**) Average profiles and heatmaps of NAB2, STAT6, and RNAPII ChIP-seq in a primary SFT at 5921 NAB2 only peaks, 718 NAB2-STAT6 peaks, and 1285 STAT6 only peaks. NAB2-STAT6 peaks had

*Figure 4 continued*

significant NAB2, STAT6, and RNAPII signal. (**c**) Top 2 motifs from motif analysis of 5921 NAB2 only peaks, 718 NAB2-STAT6 peaks, and 1285 STAT6 only peaks using HOMER shows EGR1 and WT1 as the most significantly enriched TF matrices at NAB2 and NAB2-STAT6 sites and GRE and STAT3 at STAT6 sites. (**d**) GSEA shows that genes nearest to NAB2-STAT6 peaks (n=718) are significantly upregulated in SFTs when compared with matching normal tissue from *Figure 1*. (**e**) Screenshot displays the promoter (highlighted in yellow) of *KLF10* that has significant NAB2, STAT6, and RNAPII localization in SFTs and in U2OS gains NAB2-STAT6 FLAG localization and has increased EGR1 localization and accessibility by ATAC-seq. (**f**) Screenshot displays an enhancer and promoter (highlighted in yellow) of *NAB2* that has significant NAB2 and STAT6 localization in SFTs and in U2OS gains NAB2-STAT6 FLAG localization and has increased EGR1 localization and accessibility by ATAC-seq.

The online version of this article includes the following figure supplement(s) for figure 4:

**Figure supplement 1.** NAB2-STAT6 localizes to EGR1 targets in primary tumors.

transcriptomes are the closest to SFTs, we performed single-sample gene set enrichment analysis (ssGSEA) probing the SFT gene signature across the entire Cancer Genome Atlas database (22,687 samples from 223 different tumors). About 30% of TCGA tumors, including most sarcoma subtypes, had a positive ssGSEA score indicating that a significant number of EGR1-driven genes are upregulated (*Figure 6a*). Strikingly, a group of neuroendocrine tumors (glioblastoma, mixed glioma, neuroblastoma, and pheochromocytoma/paraganglioma) stood out as the most closely correlated to SFTs (*Figure 6a*). Conversely, myeloid and lymphoid malignancies showed the poorest correlation (*Figure 6a*). Neuroendocrine tumors are broadly characterized by nerve cell traits as well as by their unique secretory features that may impact distal organs and tissues. In addition to IGF2 and IGF1 hormones, as we previously described, SFTs upregulate secreted metabolic regulators such as CTRP11, neuropeptides and their activating enzymes (NPW, PCSK2), and chemokines/cytokines such as PDGFD, NTN3, DEFB136 (*Supplementary file 1*). Neuroendocrine tumors are challenging to treat, presenting highly variable degrees of aggressiveness. We asked whether the SFT gene signature correlated with prognosis by performing a survival analysis across all TCGA samples. SFT-like tumors showed significantly worse outcomes, suggesting that expression of an EGR1 gene signature underlies tumor aggressiveness and/or therapy resistance (*Figure 6b*). We next wanted to validate the SFT signature for its ability to classify solitary fibrous tumors a priori, reasoning that ssGSEA scores should pinpoint misclassified SFTs within a pool of different tumors. Prior to NAB2-STAT6 discovery and its diagnostic use, some SFTs were misclassified as mesotheliomas due to their pleural localization (*Vejvodova et al., 2017*). We, therefore, analyzed all transcriptomes from the TCGA collection of 87 mesotheliomas and identified one patient presenting an outlier ssGSEA positive score (*Figure 6c*). We used Arriba to confirm that SH-A7BH was, in fact, the only tumor in the mesothelioma cohort containing a NAB2-STAT6 fusion (exons 1–6 of NAB2 and 16–22 of STAT6, *Figure 6d*).

The previously unrecognized neuroendocrine signature in SFTs and the startling number of neural genes activated via NAB2-STAT6/EGR1 elicits further questions about the identity of the tumor-initiating cells. To address this question, we first investigated the correlation of SFTs to normal tissue types by calculating the Spearman correlation with all samples from the Human Protein Atlas. SFTs correlated best with several neuronal tissue types and were the least correlated with lung and gut tissues (*Figure 6—figure supplement 1*). In addition to neuronal genes, we noticed that some of the top upregulated genes were implicated in embryogenesis and early development (i.e. shh regulators such as GLI2 or wnt pathway activators such as WNT2). We resolved to examine the protein domains that were enriched within the top 400 genes of the SFT signature using the functional annotation tool InterPro. The most significantly enriched domain was the Homeobox domain (*Figure 6e*), which functions as a DNA-binding domain for a large class of transcription factors that are expressed during embryonic and fetal development to drive pattern formation, axis specification, ultimately leading to proper tissue and organ morphogenesis. Notably, several homeodomain transcription factors were overexpressed in SFTs (ALX4, SHOX2, SIX1) and entire HOX clusters (HOXA, HOXC) were upregulated, as further confirmed by RNAPII profiling of a primary tumor tissue (*Figure 6f*). However, we did not identify a homeobox binding motif in our previous analysis of NAB2-STAT6 binding sites (*Figures 3b and 4c*). We then searched for enriched TF binding sites using all RNAPII-bound enhancers (excluding NAB2-STAT6 targeted enhancers) and found an overwhelming enrichment for homeotic proteins as well as Nanog (*Figure 6g*). These results suggest that a homeobox TF network, elicited by NAB2-STAT6 upregulation of homeotic genes, drives a significant part of the SFT transcriptomes.

A
**U2OS Nuclear Extract (1 day Dox)**

B

C

D

**Figure 5.** NAB2-STAT6 interacts with EGR1 and NAB1 directs them to the nucleus. (**a**) Eluates from NAB2 and STAT6 IPs from U2OS nuclear extracts expressing NAB2-STAT6 for 1 day were subjected to MudPIT LC-MS/MS analysis for unbiased identification of the top interactors. Log2 iBAQ protein scores of STAT6 IP interactors are plotted against scores of NAB IP. NAB1 and EGR1 were the top interactors. (**b**) Control and U2OS cells expressing NAB2-STAT6 for 1 day were subcellularly fractionated into nuclear and cytoplasmic fractions. Immunoblot analysis shows that NAB2-STAT6 was only

*Figure 5 continued on next page*

*Figure 5 continued*

present in the nuclear fraction of Dox conditions. STAT6 was nuclear in both conditions. NAB2 and EGR1 were cytoplasmic in control conditions but became nuclear in Dox conditions. GAPDH was cytoplasmic control and Histone H3 was nuclear control. (**c**) Immunocytochemistry (ICC) of NAB2 (red), STAT6 (green), and DAPI (blue) in solitary fibrous tumor (SFT) primary cells from *Figure 4* and U2OS control and NAB2-STAT6 (Dox) expressing for one day cells. SFT and NAB2-STAT6 expressing cells show strong nuclear staining for NAB2 and STAT6. Control U2OS cells have nuclear STAT6 and cytoplasmic NAB2 staining. (**d**) Immunocytochemistry (ICC) of NAB1 (red), FLAG (green), and DAPI (blue) in U2OS control and NAB2-STAT6 (Dox) expressing cells for 1 day. NAB2-STAT6 expressing cells show strong nuclear staining for FLAG and NAB1. Control U2OS cells have no FLAG and cytoplasmic and nuclear NAB1 staining.

The online version of this article includes the following source data and figure supplement(s) for figure 5:

**Source data 1.** Immunoblot scans with captions.

**Source data 2.** Raw immunoblot scans for *Figure 5*.

**Figure supplement 1.** NAB2-STAT6 interacts with EGR1 and NAB1 directs them to the nucleus.

## Discussion

Fusion oncoproteins frequently occur in a broad range of tumors, such as AML, sarcomas, non-small cell lung cancer, and prostate cancer (*Gao et al., 2018*). The loci encoding fusion protein products originate via chromosomal aberrations such as translocations or, as in the case of SFTs, inversions (*Mertens et al., 2015*). In most cases, the aberrant protein product retains select biological activities from both protein partners and drives neoplastic transformation (*Shen and Vakoc, 2015*; *Bushweller, 2019*; *Latysheva and Babu, 2016*).

In this work, we characterize the mechanistic role of the NAB2-STAT6 fusion in Solitary Fibrous Tumors through a combination of genomics data generated from primary SFTs, a tet-inducible model system in human cell lines, and by comparing SFT gene signatures to the Cancer Genome Atlas collection. Detection of NAB2-STAT6 protein products has become the primary diagnostic tool for this rare tumor type, however the role of the fusion protein in the etiology of SFTs has remained elusive (*Schweizer et al., 2013*). A variety of mechanisms have been proposed for NAB2-STAT6's function, including activation of STAT6 targets and conversion of NAB2 from a repressor to an activator (*Robinson et al., 2013*; *Park et al., 2019*; *Park et al., 2020*; *Li et al., 2023*). Here, we show that NAB2-STAT6 activates an EGR1-driven neuroendocrine gene expression signature by translocating EGR1, NAB1, and wild-type NAB2 to the nucleus (*Figure 7*). Our data suggest that the NAB2 moiety of the fusion protein targets a specific set of EGR1-dependent enhancers and proximal promoter sites and uses NAB1, or endogenous NAB2, as co-activators. In fact, our data suggest that NAB2/NAB1 are co-activators of EGR-1 targets under physiological conditions, however, cytoplasmic localization restrains their activity. Unlike previously proposed (*Schweizer et al., 2013*), we find that the STAT6 moiety of the fusion is the major driver of its nuclear localization (*Figure 7*). STAT6, however, does not endow the fusion protein with the ability to recognize a subset of STAT motifs, but may recruit additional co-activators. NAB1/NAB2 were originally proposed to act as co-repressor of EGR1 targets on the basis of transactivation assays (*Kumbrink et al., 2005*). Data from this work and our previous analysis of NAB2 activity during myeloid differentiation establish that these proteins operate as robust co-activators of EGR1 targets, at enhancers and proximal promoters (*Barbieri et al., 2018*). In myeloid cells, we found that full-length NAB2 recruits the Integrator protein complex. The NAB2 moiety of NAB2-STAT6 does not appear to recruit Integrator subunits, yet retains co-activator abilities by partnering with chromatin modifiers (CBP/P300) and helicases (RHA).

For the first time, we were able to establish transcriptional changes that occur in SFTs through comparison with the adjacent matching normal tissue. This led to identifying a robust EGR-1 driven neuronal gene signature, including GABA receptor subunits, synapses modulators, ion channels, and modulators of axonal morphology. In the CNS, EGR1 is known to regulate a variety of neuronal-dependent processes such as memory and behavior. We also identified a distinct secretory gene signature associated with SFTs. In fact, IGF2 is the most upregulated gene, via activation of an intronic enhancer by EGR1. IGF2 was pinpointed as the cause of hypoglycemia occurring in a very small subset of SFTs (Doege–Potter syndrome) (*Chen et al., 2021*). Our data suggest that IGF2 (and IGF1) upregulation is a common feature of all SFTs. In addition to insulin-like growth factors, STFs may secrete a host of peptides with diverse functions in neuronal processes, chemotaxis, and growth stimulation. The previously unrecognized neuronal features and the putative secretory phenotype of STFs set



**Figure 6.** The solitary fibrous tumor (SFT) gene signature resembles EGR1-activated tumors. (**a**) Ranking of TCGA tumors by their average single sample gene set enrichment analysis (ssGSEA) score using the SFT gene signature of upregulated genes from **Figure 1**. (n=2429). Neuroendocrine tumors highly express the signature while leukemias downregulate the signature. (**b**) Kaplan-Meier curve showing survival analysis of tumors in the TCGA database stratified by high or low expression of SFT gene signature of upregulated genes from **Figure 1** (n=2429). (**c**) ssGSEA of 87 mesotheliomas

*Figure 6 continued on next page*

from TCGA using SFT gene signature of upregulated genes from **Figure 1** (n=2429). Shows significant upregulation of the SFT gene signature in the TCGA SH A7BH sample. (**d**) NAB2-STAT6 gene fusion in (NAB2 exons 1–6, STAT6 exons 17–22) present in TCGA SH A7BH, originally diagnosed mesothelioma. Graphic generated with Arriba, a fusion detection algorithm. (**e**) InterPro domain analysis of the top 400 most upregulated genes in SFTs shows Homeobox and Cadherin as the most significantly enriched protein domains. (**f**) Screenshot displays HOXA locus which has significant RNAPII localization in SFTs indicating that the Homeobox genes are actively transcribed in SFTs. (**g**) Motif analysis of 11155 distal enhancer (1>kb from nearest TSS) RNAPII peaks using HOMER shows HOX, GRE, and PGR as the most significantly enriched TF matrices.

The online version of this article includes the following figure supplement(s) for figure 6:

**Figure supplement 1.** The solitary fibrous tumor (SFT) gene signature resembles EGR1-activated tumors.

them apart from mesenchymal malignancies and relate them to neuroendocrine malignancies such as pheochromocytoma, oligodendroglioma, and neuroblastoma. Neuroendocrine tumors originate from neuron-like cells that are able to send and receive signals from the nervous system, but also function as endocrine organs by producing hormones such as IGF2, with a systemic effect across distal tissues and organs. Similar to many neuroendocrine neoplasms, SFTs are immunologically cold and downregulate a set of immune response genes, perhaps with further support from IGF2 itself (**Belfiore et al., 2023**).

SFTs were classified as mesenchymal on the basis of their histological patterns and are believed to originate from soft tissue cell types, such as fibroblasts (**Jo and Demicco, 2022**). When comparing SFT signatures to human tissue RNA-seq datasets, we identified greater similarities to CNS tissues but not mesenchymal tissue types. Furthermore, we unveiled a set of embryonically and fetally expressed transcription factors that appear to coordinate part of SFT transcriptomes. Ectopic expression of a few homeotic genes has been observed across many tumors and may contribute to their growth and plasticity (**Brotto et al., 2020**). SFTs appear to upregulate an overwhelming number of homeotic transcription factors, as we showed, perhaps suggesting that the tumor mass may originate from a corrupted developmental process from embryogenesis or fetal development.

# Methods

**Key resources table**

| Reagent type (species) or resource | Designation | Source or reference | Identifiers | Additional information |
|---|---|---|---|---|
| Antibody | NAB2 | Thermo Fisher | Cat#PA5-27925 | WB, 1:1000, IF, 1:500, ChIP |
| Antibody | STAT6 | Santa Cruz Biotechnology | Cat#sc-374021 | WB, 1:1000, IF, 1:500, ChIP |
| Antibody | EGR1 | Bethyl | Cat#A303-390A | WB, 1:1000, ChIP |
| Antibody | Rabbit polyclonal anti-GAPDH | Cell Signaling | Cat#2118 | WB, 1:10,000 |
| Antibody | Mouse monoclonal anti-FLAG | Sigma | Cat#F1804 | WB, 1:5000, IF 1:500, ChIP |
| Antibody | Rabbit polyclonal anti-H3K27ac | Abcam | Cat#ab4279 | ChIP |
| Antibody | Rabbit polyclonal anti-H3K4me1 | Abcam | Cat#ab8895 | ChIP |
| Antibody | Rabbit polyclonal anti-NAB1 | Protein Tech | 18541–1-AP | IF, 1:500 |
| Antibody | Rabbit polyclonal anti-RNAPII | **Barbieri et al., 2018** | Custom made | ChIP |

*Continued on next page*

*Continued*

| Reagent type (species) or resource | Designation | Source or reference | Identifiers | Additional information |
|---|---|---|---|---|
| Antibody | Anti-rabbit IgG (HRP) | Cell Signaling | Cat#7074 | WB, 1:10,000 |
| Antibody | Anti-mouse IgG (HRP) | Cell Signaling | Cat#7076 | WB, 1:10,000 |
| Antibody | Goat anti-Rabbit IgG (H+L) Cross-Adsorbed Secondary Antibody, Alexa Fluor 594 | Thermo Fisher Scientific | Cat#A-11012 | IF, 1:500 |
| Antibody | Goat anti-Rabbit IgG (H+L) Cross-Adsorbed Secondary Antibody, Alexa Fluor 488 | Thermo Fisher Scientific | Cat#A-11008 | IF, 1:500 |
| Commercial assay or kit | Gibson Assembly Master Mix | New Englands Biolabs | Cat#E2611S | |
| Commercial assay or kit | Phusion High-fidelity DNA polymerase kit | New Englands Biolabs | Cat#M0530S | |
| Commercial assay or kit | Direct-zol RNA Miniprep kit | Zymo Research | Cat#R2051 | |
| Commercial assay or kit | Rvertaid first strand cDNA synthesis kit | Thermo Fisher Scientific | Cat#K1622 | |
| Commercial assay or kit | ChIP DNA Clean & Concentrator kit | Zymo Research | Cat#D5201 | |
| Commercial assay or kit | NEBNext Ultra II DNA Library Prep Kit for Illumina | New Englands Biolabs | Cat#E7645S | |
| Commercial assay or kit | TruSeq RNA Library Prep for Enrichment TruSeq | Illumina | Cat#20020189 | |
| Commercial assay or kit | TruSeq RNA Enrichment | Illumina | Cat#20020490 | |
| Commercial assay or kit | TruSeq RNA Enrichment | Illumina | Cat#20020490 | |
| Commercial assay or kit | TruSeq RNA Single Indexes Set A | Illumina | Cat#20020492 | |
| Commercial assay or kit | Illumina Exome Panel | Illumina | Cat#20020183 | |
| Commercial assay or kit | NEBNext Multiplex Oligos for Illumina (Index Primers set 1) | New Englands Biolabs | Cat#E6310S | |
| Commercial assay or kit | RNeasy DSP FFPE Kit | Qiagen | Cat#73604 | |
| Commercial assay or kit | Illumina Stranded Total RNA Prep, Ligation with Ribo-Zero Plus | Illumina | Cat#20040525 | |
| Other | Dynabeads Protein A | Invitrogen | Cat#10002D | |
| Other | Dynabeads Protein G | Invitrogen | Cat#10004D | |
| Other | IgG elution buffer | G-Biosciences | Cat#786–545 | |
| Other | Anti-FLAG M2 Magnetic Beads | Sigma-Aldrich | Cat#M8823 | |
| Other | FLAG peptide | Sigma-Aldrich | Cat#F3299 | |

*Continued on next page*

*Continued*

| Reagent type (species) or resource | Designation | Source or reference | Identifiers | Additional information |
|---|---|---|---|---|
| Cell line (*Homo sapiens*) | 293T cells | ATCC | Cat#CRL-3216 | |
| Cell line (*Homo sapiens*) | U2OS | Lakadamyali Lab, University of Pennsylvania | | |
| Peptide, recombinant protein | FLAG peptide | Sigma-Aldrich | Cat#F3299 | |
| Sequence-based reagent | pINTO_INTS13-F | This paper | Gibson assembly cloning | AACCGGTAGCGGTAC C AAGATTTTTTCTGAATCT |
| Sequence-based reagent | pINTO_INTS13-R | This paper | Gibson assembly cloning | ATCCGAGCTCGGTACCTCAC TGCCGGCTGGCTTT |
| Sequence-based reagent | pINTO_INTS14-F | This paper | Gibson assembly cloning | AACCGGTAGCGGTAC C CCGACAGTGGTGGTAATG |
| Sequence-based reagent | pINTO_INTS14-R | This paper | Gibson assembly cloning | ATCCGAGCTCGGTACCTCAAAT TCTTTCAGTGCT |
| Sequence-based reagent | pINTO_INTS10-F | This paper | Gibson assembly cloning | AACCGGTAGCGGTAC C TCTGCCCAGGGGGACTGC |
| Sequence-based reagent | pINTO_INTS10-R | This paper | Gibson assembly cloning | ATCCGAGCTCGGTACC TCAGGTCAGAGTCTGAAG |
| Sequence-based reagent | INTS10-F | This paper | PCR primers | TATACCAGTGGCCTCTTGTC |
| Sequence-based reagent | INTS10-R | This paper | PCR primers | CGTCTTCCTTTATCCATCTGCC |
| Recombinant DNA reagent | pLVX-Tight-Puro | Clontech | | |
| Recombinant DNA reagent | pLVX-Tet-On Advanced | Clontech | | |
| Recombinant DNA reagent | pLENTI-CMV-GFP-Puro | Addgene | | |
| Recombinant DNA reagent | Tet-pLKO-puro | Addgene | | |
| Recombinant DNA reagent | pLVX-NAB2-STAT6-FLAG-Tight-Puro | This paper | | |
| Recombinant DNA reagent | pLenti-NAB2-STAT6-FLAG-puro | This paper | | |
| Software, algorithm | STAR (2.5) | *Dobin et al., 2013* | | |
| Software, algorithm | Samtools (0.1.19) | *Danecek et al., 2021* | | |
| Software, algorithm | FeatureCounts | *Liao et al., 2014* | | |
| Software, algorithm | DESeq2 (1.38.3) | *Love et al., 2014* | | |
| Software, algorithm | MACS2 | *Zhang et al., 2008* | | |
| Software, algorithm | bedtools (2.26.0) | *Quinlan and Hall, 2010* | | |

*Continued on next page*

*Continued*

| Reagent type (species) or resource | Designation | Source or reference | Identifiers | Additional information |
|---|---|---|---|---|
| Software, algorithm | ggplot | *Hamilton and Ferry, 2018* | | |
| Software, algorithm | deeptools (3.5.1) | *Ramírez et al., 2016* | | |
| Software, algorithm | DiffBind (3.4.11) | *Stark and Brown, 2011* | | |
| Software, algorithm | gprofiler2 (0.2.1) | *Kolberg et al., 2020* | | |
| Software, algorithm | Survival (3.5–5) | *Therneau and Lumley, 2019* | | |
| Software, algorithm | GSVA (1.48.1) | *Hänzelmann et al., 2013* | | |
| Software, algorithm | HOMER (4.10.1) | *Heinz et al., 2010* | | |

## FFPE RNA-seq

FFPE sections were obtained from the Tumor Tissue and Biospecimen Bank at the Hospital of the University of Pennsylvania. Total RNA was purified using the RNeasy DSP FFPE Kit (73604) following the manufacturer's manual. Libraries were generated using the TruSeq RNA Library Prep for Enrichment (20020189) with the TruSeq RNA Enrichment (20020490), TruSeq RNA Single Indexes Set A (20020492), and the Illumina Exome Panel (20020183). Libraries were sequenced on Illumina NextSeq 2000 with 40 base pair paired-end reads. Reads were aligned to hg19 human reference genome using STAR v2.5. FeatureCounts was used for counting reads to the genes. Data were normalized using Voom and differential gene expression analysis was performed using DESeq2 in R (v1.38.3) unless otherwise noted. Data was visualized using ggplot2 (3.3.6). GO enrichment analysis was done using gprofiler2 package in R (v 0.2.1). Gene set enrichment analysis (GSEA) was 500 randomly selected genes from the selected data sets using the clusterProfiler package in R (v4.6.2).

## Cell culture

293T were purchased from American Type Culture Collection (ATCC) and maintained in Dulbecco's Modified Eagle's Medium (DMEM) supplemented with 10% super calf serum and Glutmax. U2OS cells were obtained from the Lakadamyali lab and maintained in Dulbecco's Modified Eagle's Medium (DMEM) supplemented with 10% tet-free FBS or regular FBS and Glutmax. Cells were fed with fresh medium every 3–4 days until reaching 80–90% confluence and split to 1:6-1:12 during passaging.

## Lentiviruses packaging

Lentiviruses were produced in HEK293T cells by co-transfection 8 µg of select plasmid with three packaging plasmids (2.5 µg of pRSV-REV, 5 µg of pMDLg/pRRE and 3 µg of pMD2.G per 10 cm cell culture dish) using calcium phosphate transfection (*Chen and Okayama, 1987*). Lentiviruses were harvested 48–72 hr after transfection. Lentivirus was used fresh or pelleted by centrifugation at 16,500 rpm for 90 min then pellets were air dried for 30 min and resuspended in 10 µl of PBS, aliquoted, and frozen in −80°C.

## Cloning

Gene blocks containing three fragments of the long isoform of NAB2-STAT6 (exons 1–4 of NAB2 and exons 2–22 of STAT6) with a C-terminal FLAG tag were ordered from GenScript. Using Gibson Assembly Master Mix (NEB, cat#E2611), gene blocks were assembled with a digested pLVX-Tight-Puro (ClonTech) or pLENTI-CMV-GFP-Puro (Addgene, cat#17448) vectors. shRNA oligos were ordered from IDT and cloned into digested Tet-pLKO-puro (21915) vector by ligation. All plasmids generated were verified by Sanger sequencing.



**Figure 7.** NAB2-STAT6 drives the expression of EGR1 targets by driving co-activators to the nucleus. Model for NAB2-STAT6's function; NAB2-STAT6 is directed to the nucleus by its STAT6 moiety. The fusion protein then directs co-activators NAB1, NAB2, and EGR1 to the nucleus which in turn directs NAB2-STAT6 to EGR1 target promoters and enhancers which are highly activated by the complex of co-activators recruited. This figure was created using BioRender.com.

### Generating tet-inducible NAB2-STAT6 system in U2OS cells

U2OS cells were plated into a 6-well plate at 30–40% confluence in media with tet-free FBS which was used for the rest of the process. When the cells reached 60–70% confluence, 2 ml of medium fresh lentivirus media generated using pLVX-Tet-On Advanced (ClonTech) with 8 µg/ml polybrene (Thermo Fisher, cat#TR1003G) per well was added to replace the old medium. 24 hr after induction, the virus medium was removed and replaced with fresh cell culture medium for another 48 hr. After that, cells were selected with 200 µg/ml of Neomycin (Corning, cat#MT30234CR) in fresh medium. Cells were selected in Neomycin for 2 weeks with fresh media being added 3–4 weeks and cells split 1:3 when reaching 80–90% confluency. Then, cells were plated into a 6-well plate at 30–40% confluence. When the cells reached 60–70% confluence, 2 ml of medium fresh lentivirus media generated using pLVX-NAB2-STAT6-FLAG-Tight-Puro with 8 µg/ml polybrene (Thermo Fisher, cat#TR1003G) per well was added to replace the old medium. 24 hr after induction, the virus medium was removed and replaced with fresh cell culture medium for another 48 hr. After that, cells were selected with 0.5 µg/ml of puromycin in fresh medium (InvivoGen, cat#ant-pr-1). 48 hr after selection with puromycin, cells were disassociated with trypsin and plated at a low density in a 15 cm dish. Single cells were cultured with 0.5 µg/ml of puromycin for the next 2–3 weeks until colonies appeared. Individual microcolonies were moved to a 96-well plate for clonal expansion. Clones were screened after addition of 1 µg/mL Doxycycline by verified by Western blot.

### Generating constitutively expressing NAB2-STAT6 in HEK293T

pLenti-NAB2-STAT6-FLAG-puro plasmid transfected to HEK293T cells using Lipofectamine 2000 (Thermo Fisher, cat#11668030) as the manufacturer's instructions. 24–48 hr after transfection, the old medium was replaced with fresh medium with 400 µg/ml zeocin (InvivoGen, cat#ant-zn-1) for 48 hr to remove non-transfected cells. After that, trypsinized single cells were plated in 15 cm dishes and maintained in fresh medium containing 400 µg/ml zeocin for 2–3 weeks for clonal growth. Expression of NAB2-STAT6 FLAG in microclones was verified by immunoblotting with an M2-Flag antibody.

### Immunoprecipitation-mass spectrometry (IP-MS)

Nuclear fractions were extracted from cells for IP experiments. Briefly, cells were collected and washed two times with cold PBS. Cell pellets were resuspended in 5 packed volumes (PCV) of Buffer A (10 mM HEPES, 5 mM MgCl2, 0.25 M Sucrose, 0.5 mM DTT, and 1 mg/ml each of protease inhibitors aprotinin, leupeptin, and pepstatin) then NP-40 was added to 0.1% concentration and cells were mixed again. Resuspended cells were incubated on ice for 10 min. Cells were then pelleted at 8000 rpm for 10 min at 4 °C. Supernatant was saved as the cytoplasmic fraction. Pellets from previous spin was resuspended in 4 PCV of Buffer B (10 mM HEPES, 1.5 mM MgCl2, 25% glycerol, 0.1 mM EDTA, 0.5 mM DTT and 1 mg/ml each of protease inhibitors aprotinin, leupeptin, and pepstatin) then NaCl was added to 0.5 M concentration and cells were mixed again. The resuspended extract was incubated for 20 min on ice. Cells were then sonicated briefly for 6 s. The extract was then pelleted at 10,000 rpm for 10 min at 4 °C. Supernatant was saved as the nuclear fraction. Nuclear and cytoplasmic fractions were further cleared at 15,000 rpm for 30 min at 4 °C. All the saved extracts were dialyzed overnight in BC80 (20 mM Tris pH 8.0, 80 mM KCl, 0.2 mM EDTA, 10% glycerol, 1 mM B-mercaptoethanol, 0.2 mM phenylmethylsulfonyl fluoride (PMSF)) at 4 °C. The extracts were spinned at 20,000 g for 60 min at 4 °C on the next day. Supernatant was used for experiments or saved at –80 °C.

For each IP, 2 mg of nuclear extract, 4 ug of antibody, 30 µl of Dynabeads Protein A or G was mixed in co-IP buffer (20 mM Tris pH 8.0, 100 mM NaCl, 0.1% NP-40, 0.5 mM DTT, and 1 mg/ml each of protease inhibitors aprotinin, leupeptin, and pepstatin) to make a 500 µl volume of reaction. IPs were incubated at 4 °C for 2 hr with rotation and then washed three times with co-IP buffer and one time with PBS with 0.05% NP-40. Proteins were eluted by IgG elution buffer and analyzed by western blot or LC-MS/MS.

For Flag IP, 2 mg of nuclear extract was incubated with 20 µl of anti-FLAG M2 Magnetic beads and eluted with FLAG peptide.

### Western blot

Cells were harvested and washed three times in 1 X PBS. Cell pellets were resuspended and lysed in cold RIPA buffer (50 mM Tris-HCl pH 7.5, 150 mM NaCl, 1% Igepal, 0.5% sodium deoxycholate, 0.1%

SDS, 500 µM DTT) supplemented with 1 µg/ml of each of the protease inhibitor aprotinin, leupeptin, and pepstatin. 20 µg of cell lysate was loaded in Bolt 4–12% Bis-Tris gel (Invitrogen) and separated through gel electrophoresis in 1 X Bolt MES running buffer. Proteins were transferred to ImmunoBlot PVDF membranes in Tris-Glycine buffer (Bio-Rad). Membranes were blocked with 10% BSA in TBST for 35 min at room temperature, then incubated with primary antibodies diluted in 5% BSA in 1 X TBST for 2 hr at room temperature or overnight at 4°C. Membranes were washed three times with TBST for 10 min and incubated with HRP-conjugated secondary antibodies for 1 hr at room temperature. Proteins were detected using Clarity Western ECL substrate (Biorad) and imaged with ImageQuant LAS 4000 (GE Healthcare).

## RNA extraction

RNA was extracted from cells using TRIzol and purified with the Zymo Directed RNA Miniprep kit (Zymo Research, R2050) following the manufacturer's instructions. Briefly, media was removed from the cells and wells were washed with PBS once. 1 ml Accutase was added to each well of a 6-well plate for 5 min. Cell suspension was collected and centrifuged at 300 g for 5 min. 300 µl TRI reagent was added to each pellet and either frozen at −80°C or proceeded with purification following the Zymo Directed protocol. RNA concentration was quantified by Nanodrop.

## 3' RNA Quant-seq

Cells were lysed with TRIzol and total RNA was purified using Zymo Research Direct-zol RNA mini-prep kit (R2050) following the manufacturer's manual. Libraries were generated using the QuantSeq 3' - mRNA Seq Library Prep Kit for Illumina (Lexogen). 75 base-pair single-end reads were sequenced on the Illumina NextSeq 2000. Reads were aligned to hg19 human reference genome using STAR v2.5. FeatureCounts (*Liao et al., 2014*) was used for counting reads to the genes. Data were normalized using Voom and differential gene expression analysis was performed using DESeq2 in R (v1.38.3) unless otherwise noted. Data was visualized using ggplot2 (3.3.6). GO enrichment analysis was done using gprofiler2 package in R (v 0.2.1). Gene set enrichment analysis (GSEA) was done with 500 randomly selected genes from the given set of genes across the C2 collection of the human molecular signatures database or custom signatures using the GSEA function in clusterProfiler package in R (v4.6.2).

## RNA-seq

Cells were lysated with TRIzol and total RNA was purified using Zymo Research Direct-zol RNA mini-prep kit (R2050) following manufacturer's manual. Libraries were generated using the Illumina Stranded Total RNA Prep with Ribo-Zero Plus (20040525). 40 base-pair single-end reads were sequenced on the Illumina NextSeq 2000. Reads were aligned to hg19 human reference genome using STAR v2.5. FeatureCounts (*Liao et al., 2014*) was used for counting reads to the genes. Data were normalized using Voom and differential gene expression analysis was performed using DESeq2 in R (v1.38.3) unless otherwise noted. Data was visualized using ggplot2 (3.3.6). GO enrichment analysis was done using gprofiler2 package in R (v 0.2.1). Gene set enrichment analysis (GSEA) was done with 500 randomly selected genes from the given set of genes across the C2 collection of the human molecular signatures database or custom signatures using the GSEA function in clusterProfiler package in R (v4.6.2). InterPro domain analysis was done using DAVID (https://davidbioinformatics.nih.gov/tools.jsp).

## Chromatin immunoprecipitation sequencing

Cells were resuspended at a concentration of 10 million cells per 10 ml in fresh media at room temperature in 50 ml falcon tubes. For each replicate, 10–20 million cells were harvested for cross-linking. The tube was then rotated on a rocker for 15 min at room temperature for 5 min with 1% of formaldehyde (Sigma; Cat#252549). To quench the cross-linking reaction, 560 µl of 2.5 M Glycine per 10 ml media was added, and cells were incubated at room temperature with rotation for another 10 min. After being washed twice with cold PBS and spun at 2500 rpm for 10 min at 4°C, then frozen at –80°C. Cells were then resuspended in ChIP lysis buffer (150 mM NaCl, 1% Triton X-100, 5 mM EDTA, 10 mM Tris-Cl, 500 uM DTT, 0.4% SDS) and sonicated to an average length of 200–250 bp using a Covaris S220 Ultrasonicator. Fragmented chromatin was cleared at 13,000 rpm for 10 min and diluted with SDS-free

ChIP lysis buffer. For each immunoprecipitation, cleared fragmented chromatin was incubated with 5 μg of human antibody, and Protein A or Protein G Dynabeads (Invitrogen) at 4°C overnight. After incubation, beads were washed twice with each of the following buffers: Mixed Micelle Buffer (150 mM NaCl, 1% Triton X-100, 0.2% SDS, 20 mM Tris-HCl (pH 8.0), 5 mM EDTA, 65% sucrose), Buffer 500 (500 mM NaCl, 1% Triton X-100, 0.1% Na-deoxycholate, 25 mM HEPES, 10 mM Tris-HCl (pH 8.0), 1 mM EDTA), LiCl/detergent wash buffer (250 mM LiCl, 0.5% Na-deoxycholate, 0.5% NP-40, 10 mM Tris-HCl (pH 8.0), 1 mM EDTA), followed by a final wash with 1 X TE. To elute samples, beads were then resuspended with 1 X TE supplemented with 1% SDS and incubated at 65°C for 10 min. After eluted twice, samples and the untreated input (5% of the total sheared chromatin) were incubated at 65°C overnight to reverse cross-link. After reverse cross-linking, samples were treated with 0.5 mg/ml proteinase K at 65°C for 1 hr and purified with Zymo ChIP DNA Clean Concentrator kit (Zymo Research D5205) as the manufacturer's manual and quantified by QUBIT. Barcoded libraries were made using NEB Ultra II DNA Library Prep Kit for Illumina following manufacturer's instructions and quantified by Agilent 2100 Bioanalyzer System. Libraries were sequenced on Illumina NextSeq 2000 with 40 base pair paired-end reads. Sequences were aligned to human reference hg19. Samtools (1.9.0) was used to remove the PCR duplicates (rmdup) and the reads with a mapping quality score of less than 10 from the aligned reads. Bigwig files of the data generated with deeptools (v2.4.2, bamCoverage–binSize 10–normalizeTo1 ×3137161264–extendReads 150–ignoreForNormalization chrX) and visualized on the WashU Epigenome Browser (https://epigenomegateway.wustl.edu/browser/) or the UCSC Genome Browser (https://genome.ucsc.edu/). For normalization of the data, each number of the filtered reads was divided by the lowest number of the filtered reads in the same set of experiments, generating a downsampling factor for each sample. Normalized BAM files were generated using samtools view -s with the above downsampling factors and further converted to normalized BAM files using bamCoverage–binSize 10–extendReads 150. Peaks were called by MACS2 (*Zhang et al., 2008*).

Heatmaps were generated from read depth normalized bigwig files using deeptools ComputeMatrix and visualized with plotHeatmap. Differential binding analysis was done using Diffbind package (v3.4.11). Motif analysis was done using HOMER's (4.10.1) findMotifsGenome command and known motifs were plotted.

## FLAG chromatin immunoprecipitation sequencing

FLAG ChIP-seq was done as above with the following modifications. M2-FLAG antibody was washed in one volume of PBST (1 X PBS +0.01% Tween) and then pre-bound with protein G Dynabeads for at least 4 hr. Fixed cells were then resuspended in Sonication buffer (1/80 volume of 20% Sarkosyl (FC 0.25%), 1 mM DTT, and protease inhibitors into RIPA Buffer 3.0 (0.1% SDS, 1% Triton X-100, 10 mM Tris-HCl (pH 7.4), 1 mM EDTA, 0.1% NaDOC, 0.3 M NaCl)). After incubation, beads were washed twice with each of the following buffers: Low Salt (150 mM NaCl, 1% Triton X-100, 0.1% SDS, 50 mM Tris-HCl (pH 8.0), 5 mM EDTA, 65% sucrose), High Salt (500 mM NaCl, 1% Triton X-100, 0.1% Na-deoxycholate, 25 mM HEPES, 10 mM Tris-HCl, 1 mM EDTA), LiCl/detergent wash buffer (150 mM LiCl, 0.5% Na-deoxycholate, 1% NP-40, 10 mM Tris-HCl (pH 8.0), 1 mM EDTA, 0.1% SDS), followed by a final wash with 1 X TE with 50 mM NaCl. To elute samples, beads were then resuspended in 210 μL Elution Buffer (1% SDS, 50 mM Tris-HCl (pH 8.0), 10 mM EDTA, 200 mM NaCl) and incubated at 65°C for 30 min shaking at 1200 rpm then spun down for 1 min at 16,000 g at RT. The supernatant was then collected, then samples and the untreated input (5% of the total sheared chromatin) were incubated at 65°C overnight to reverse cross-link.

## Omni-ATAC-seq

Omni ATAC-seq was performed according to Corces et al. 2017. Briefly, 50,000 cells were washed in ATAC-resuspension buffer (RSB) with 0.1% Tween-20 and then lysed by incubating on ice for 3 min in RSB with 0.1% NP-40 and 0.1% Tween-20. Lysis was washed out in RSB 0.1% Tween-20, and nuclei were pelleted by centrifugation at 600 g/4 °C/5 min. Nuclei were then incubated in the transposition mixture for 30 min at 37 °C while shaking at 1000 rpm. DNA was then purified using Zymo DNA clean and concentrator-5 Kit (cat# D4014). DNA was then amplified for five cycles with ATAC index primers and NEBNext Ultra II Q5 Master Mix (NEB #M0544). Additional amplification cycles were then determined by qPCR using 5 μL of original amplification with PerfeCTa SYBR Green FastMix Reaction Mixes

(Quantabio 95072–012). After additional amplification cycles with remaining 15 µL DNA was then purified using Zymo DNA clean and concentrator-5 Kit (cat# D4014) and quantified by QUBIT.

## Immunocytochemistry (ICC)

Immunofluorescence experiments were performed as follows *Poirier et al., 2008*. Briefly, cells were fixed with 4% formaldehyde for 15 min at room temperature, washed three times with PBS for 15 min and incubated with 1% goat serum in PBST containing 0.1% Triton X-100 for 30 min at room temperature. Cells were incubated with primary antibodies at 4°C for overnight, washed three times with PBS for 10 min at room temperature and incubated with secondary antibodies for 1 hr at room temperature. After that, cells were incubated with 1 µg/ml of DAPI for 15 min and mounted in SlowFade Gold Antifade Mountant (Thermo Fisher Scientific, Cat#S36938) and imaged using Nikon 80i Upright Microscope.

## Public data processing

Gene expression counts from TCGA were downloaded form the GDC data portal (https://www.ncbi.nlm.nih.gov/projects/gap/cgi-bin/study.cgi?study_id=phs000567.v1.p1). Publicaly available SFT RNA-seq was downloaded from the database of Genotypes and Phenotypes (dbGaP) under accession phs000567.v1.p1. Single sample gene set enrichment analysis (ssGSEA) was done using using the GSVA (1.48.1) package in R. 500 randomly selected genes from curated data sets and the TCGA, SFT FFPE RNA-seq samples, and phs000567.v1.p1 SFT RNA-seq. Survival Analysis was done using the Survival (3.5–5) package in R. Spearman correlation of SFTs and human tissues and cells was performed on the Human Protein Atlas RNA-seq datasets (https://www.proteinatlas.org/about/download#:~:text=rna_tissue_consensus.tsv.zip and https://www.proteinatlas.org/about/download#:~:text=rna_single_cell_type.tsv.zip).

## Acknowledgements

This study was supported by grants from the G Harold and Leila Y Mathers Charitable Foundation (AG) and the NIH (R01 HL141326 and R01 CA252223). CMH was also supported by a Ruth L Kirschstein NRSA Award F31 CA265257. We thank the Sarma lab for reagents and useful discussions. We thank the Tumor Tissue and Biospecimen Bank and we also thank the Genomics Core, the Proteomics & Metabolomics Core, and the Imaging Facility of The Wistar Institute (P30-CA010815) for providing outstanding technical support. Primary SFT tissue was collected in the context of an IRB-approved protocol (Penn IRB #843351). We thank the University of Pennsylvania's Tumor Tissue and Biospecimen Bank (TTAB) for help in providing primary samples. Illustrations were generated with BioRender.com.

# Additional information

## Funding

| Funder | Grant reference number | Author |
| --- | --- | --- |
| National Heart, Lung, and Blood Institute | HL141326 | Alessandro Gardini |
| National Cancer Institute | CA252223 | Alessandro Gardini |
| National Cancer Institute | CA265257 | Connor Hill |

The funders had no role in study design, data collection and interpretation, or the decision to submit the work for publication.

## Author contributions

Connor Hill, Conceptualization, Data curation, Formal analysis, Investigation, Visualization, Writing – original draft; Alexandra Indeglia, Francis Picone, Cara Cipriano, Investigation; Maureen E Murphy, Conceptualization, Investigation; Robert G Maki, Investigation, Writing – review and editing;

Alessandro Gardini, Conceptualization, Writing – original draft, Project administration, Writing – review and editing

**Author ORCIDs**
Maureen E Murphy ⬤ https://orcid.org/0000-0001-7644-7296
Alessandro Gardini ⬤ https://orcid.org/0000-0003-0413-2855

Joint Public Review: https://doi.org/10.7554/eLife.98072.3.sa1
Author response https://doi.org/10.7554/eLife.98072.3.sa2

## Additional files

### Supplementary files

Supplementary file 1. Solitary fibrous tumors express a neuronal gene signature. Table of differentially expressed genes in solitary fibrous tumors (SFTs) versus normal matching tissues as determined by formalin-fixed paraffin-embedded (FFPE) RNA-seq (n=8). 2429 genes were upregulated (indicated by red dots) and 3769 genes were downregulated. Fold change >1, FDR <0.1.

Supplementary file 2. Generation of an inducible NAB2-STAT6 system to investigate early transcriptional changes. Table of differentially expressed genes in cells expressing NAB2-STAT6 (Dox) for 2 days versus control cells as determined by 3' mRNA Quant-seq (n=4). 562 genes were upregulated and 211 genes were downregulated. Fold change >1, FDR <0.1.

Supplementary file 3. EGR1 targeted promoters and enhancers are activated by NAB2-STAT6. Table of 1394 NAB2-STAT6 FLAG peaks in 2 days NAB2-STAT6 (Dox) expressing U2OS cells annotated to the nearest TSS and classified as either enhancer (greater than 1 kb to nearest TTS) or promoter peaks.

Supplementary file 4. NAB2-STAT6 localizes to EGR1 targets in primary tumors. Table of 718 NAB2-STAT6 peaks (overlapping in NAB2 and STAT6 ChIP-seq experiments) in a primary solitary fibrous tumor (SFT) annotated to the nearest TSS and classified as either enhancer (greater than 1 kb to nearest TTS) or promoter peaks.

MDAR checklist

### Data availability

Sequencing data have been deposited in GEO under accession code GSE249703.

The following dataset was generated:

| Author(s) | Year | Dataset title | Dataset URL | Database and Identifier |
|---|---|---|---|---|
| Hill CM, Indeglia A, Picone F, Murphy M, Cipriano C, Maki RG, Gardini A | 2024 | NAB2-STAT6 drives an EGR1-dependent neuroendocrine program in Solitary Fibrous Tumors | https://www.ncbi.nlm.nih.gov/geo/query/acc.cgi?acc=GSE249703 | NCBI Gene Expression Omnibus, GSE249703 |

The following previously published dataset was used:

| Author(s) | Year | Dataset title | Dataset URL | Database and Identifier |
|---|---|---|---|---|
| NCI | 2013 | Identification of recurrent NAB2-STAT6 gene fusions in solitary fibrous tumor by integrative sequencing | https://www.ncbi.nlm.nih.gov/projects/gap/cgi-bin/study.cgi?study_id=phs000567.v1.p1 | NCBI dbGaP, phs000567.v1.p1 |

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
