## [Editor Report · eLife Assessment]

This study provides **compelling** data regarding the molecular characterization of a rare tumor type with few treatment options. This **fundamental** work significantly advances our mechanistic understanding of solitary fibrous tumours, a critical first step towards targeted precision medicine approaches. The results of this study will be of broad interest to cancer biologists and experimental oncologists.

---

## [Referee Report · Joint Public Review]

Solitary Fibrous Tumors (SFTs) are a rare malignancy defined by NAB2-STAT6 fusions. Because the molecular understanding of the disease is largely lacking, there are currently no targeted treatment approaches. Using primary tumor and adjacent normal tissue samples and cells inducibly expressing NAB2-STAT6, Hill et al. perform a detailed characterization of the transcriptomic and epigenomic NAB2-STAT6 SFT signatures. They identify enrichment or EGR1/NAB2 (but not STAT6) sites bound by the fusion protein and increased expression of EGR1 targets. Their studies indicate that NAB2-STAT6 fusion may direct the nuclear translocation of NAB2 and EGR1 proteins and potentially NAB1. Transcriptionally, NAB2-STAT6 SFTs most closely resemble neuroendocrine tumors.

This pioneering study provides critical insight into the molecular pathogenesis of SFTs, pivotal for the future development of mechanistically informed treatment approaches. The study is rigorously executed and well-written. This new knowledge is an important addition to the field.

---

## [Author Response]

The following is the authors’ response to the original reviews.

**Response to the Joint Public Review:**

We are indebted to eLife’s reviewing process for helping us improve our manuscript and for highlighting that our study provides new molecular insights into SFT pathogenesis.

**Response to Reviewers:**

(1) The authors state that "NAB2-STAT6 localization is exclusively driven by EGR1 binding" yet WT1 motives are also consistently enriched. Can you please touch upon the potential involvement of WT1 (or lack thereof, and why)?

Our data suggest that EGR1 is the primary driver of NAB2-STAT6 localization. In fact, EGR1 is the most significantly enriched motif (Fig. 4) at NAB2-STAT6 binding sites and we detect an interaction between the fusion protein and EGR1 (Fig. 5). Conversely, we did not identify an interaction between NAB2-STAT6 and WT1. However, WT1 also belongs to the C2H2 zinc finger subclass and recognizes a motif bearing striking similarities to the EGR1/2 consensus. EGR1 has been previously described to bind WT1 motifs and to function as an activator of WT1 targets (as opposed to WT1 repressive abilities). See https://www.jbc.org/article/S0021-9258(20)74720-4/fulltext and https://www.sciencedirect.com/science/article/pii/S0378111901005935.

(2) In the description of Figure 5C the authors observe nuclear staining of both NAB2 and STAT6 following NAB2-STAT6 fusion induction. They interpret this as the fusion stimulates nuclear translocation of endogenous NAB2. This statement can only be rigorously made if the authors can unequivocally demonstrate that their antibody exclusively detects endogenous NAB2 and not the NAB2 portion of the fusion. As presented, a more likely interpretation is that the NAB2 staining detects NAB2-STAT6 fusion protein. Since there is some cytoplasmic NAB2 signal still present, the findings in Figure 5c do not support nor disprove nuclear translocation of endogenous NAB2. It may be prudent to remove this section. Figure 5B is currently the best direct evidence of nuclear translocation.

We agree with the reviewer that Fig. 5C does not rigorously show that NAB2-STAT6 fusion proteins drag endogenous NAB2 into the nucleus. The immunostaining reveals that wt NAB2 localization is overwhelmingly cytoplasmic at steady-state conditions (and prior to expression of the fusion protein). Instead, Figure 5B shows that endogenous NAB2 translocates to the nucleus upon NAB2-STAT6 expression. Additionally, figure 5A (along with Suppl. Fig. 5 E-F) demonstrates that endogenous NAB2 co-precipitates with NAB2-STAT6 fusions in nuclear extracts of U2OS and HEK293T cells. We have rephrased the paragraph accordingly.

(3) Figure 5D: for the interpretation of the presented data to hold up, namely, NAB1 nuclear translocation upon NAB2-STAT6 expression, it is important to demonstrate that NAB1 antibodies do not cross-react with NAB2 given the similarity between NAB1 and NAB2. Without such control, another likely interpretation of the results in Figure 5D is that NAB1 antibody detects the NAB2 portion of the overexpressed fusion protein. This needs to be acknowledged in the text.

We had similar concerns, therefore we confirmed that the NAB1 antibody does not cross react with NAB2 by immunoblot (see figure below). We overexpressed FLAG-NAB2, HA-NAB1 and GFP constructs in HEK293T cells, we performed immunoprecipitation with either HA or FLAG from whole cell extracts followed by western blot using anti-NAB2 and anti-NAB1 polyclonal antibodies. We did not observe cross-reactivity of these antibodies. We acknowledged antibody validation in the revised text.

**Author response image 1. sa2fig1:** 

(4) Also, to support the notion that NAB2-STAT6 fusion promotes nuclear translocation of the entire complex, an imaging approach detecting EGR1 similar to Figure 5C-D would be helpful. EGR1 staining also avoids the potential pitfall of NAB1/2 antibodies detecting NAB2-STAT6 overexpressed fusion instead of endogenous proteins.

We agree with the reviewer that this would be a helpful approach. Unfortunately, none of the commercially available EGR1 antibodies that we tested were suitable for immunocytochemistry, as they either failed to show a proper signal or were marred by high nonspecific background signal.

(5) The authors found increased mRNA expression of certain cytokines and secreted neuropeptides in SFTs. While this may be consistent with a secretory phenotype, additional evidence such as detection of elevated levels of these proteins in tumor lysates or in culture media is necessary to formally make this claim. Please rephrase.

We have rephrased our claims as suggested. The revised text is now as follows: “We also identified a distinct secretory gene signature associated with SFTs. In fact, IGF2 is the most upregulated gene, via activation of an intronic enhancer by EGR1. IGF2 was pinpointed as the cause of hypoglycemia occurring in a very small subset of SFTs (Doege–Potter syndrome)(52). Our data suggest that IGF2 (and IGF1) upregulation is a common feature of all SFTs. In addition to insulin-like growth factors, STFs may secrete a host of peptides with diverse functions in neuronal processes, chemotaxis, and growth stimulation. The previously unrecognized neuronal features and the putative secretory phenotype of STFs set them apart from mesenchymal malignancies and relate them to neuroendocrine malignancies such as pheochromocytoma, oligodendroglioma and neuroblastoma.”

(6) GSEA with 500 randomly selected genes from target datasets needs a more detailed description to clarify the method.

To improve clarity, we added the following description: “Gene set enrichment analysis (GSEA) was done with 500 randomly selected genes from the given set of genes across the C2 collection of the human molecular signatures database or custom signatures using the GSEA function in clusterProfiler package in R (v4.6.2).

(7) In the IP-MS description, please double check the NaCl concentration in the second extraction step - 0.5mM seems low. Also, in the IP part, a buffer recipe appears to have been incorrectly pasted.

We thank the reviewer for identifying this typo. Indeed, we used 0.5M NaCl instead of 0.5mM. We have corrected the co-IP buffer recipe accordingly.